# The Perspectives of Hydrophobic Coatings for Mitigating Icing on Atmospheric Structures

Xingliang Jiang [1,*], Yizhi Zhuo [2], Peng Wang [3], Mengyu Yang [3], Yongli Liao [4] and Baohui Chen [5]

1 Xuefeng Mountain Energy Equipment Safety National Observation and Research Station of Chongqing University, Chongqing 400044, China
2 Institute of Solid State Physics, Hefei Institutes of Physical Science, Chinese Academy of Sciences, Hefei 230031, China
3 Department of Mechanical Engineering, North China Electric Power University, Baoding 071066, China
4 Electric Power Research Institute, China Southern Power Grid Corporation, Guangzhou 510630, China
5 State Key Laboratory of Disaster Prevention & Reduction for Power Grid Transmission and Distribution Equipment, State Grid Corporation of China, Changsha 410023, China
* Correspondence: xljiang@cqu.edu.cn

**Abstract:** Ice accumulation on atmospheric structures will result not only in inconvenience to human activities, but also various catastrophic events. Many anti-icing coatings have been developed for anti-ice accretion on various atmospheric structures. However, such mitigating icing performances and developments in practical applications are restricted by various factors. Therefore, current mitigating icing coatings are far from practical implementation. Rough and smooth hydrophobic coatings have demonstrated their potential for mitigating ice formation. To advance the development of mitigating icing coatings, a perspective of hydrophobic coatings for mitigating icing is in need. Herein, this paper categorizes the mitigating icing coatings by their wettability firstly. Then, we recap the state-of-the-art hydrophobic coatings for mitigating icing. Afterwards, we point out the deficiency and limitations of current coatings for anti-icing. At last, we provide a perspective of future trends and development directions. This perspective review can guide the design of mitigating icing coatings towards practical implementation.

**Keywords:** hydrophobic; perspectives; mitigating icing; icephobic; ice adhesion

## 1. Introduction

Ice accretion can occur on the surfaces of atmospheric structures, such as power lines, airplanes, solar panels, wind turbines and other infrastructures and transportation. The accumulation of ice on various atmospheric structures tends to bring inconvenience to human activities. In some cases, it may even cause catastrophic events, such as plane crashes and power network outages [1,2]. Among them, ice-covered transmission lines may cause ice flash tripping and power lines' violent dancing, then leading to lines breaking and tower falling accidents [3,4]. In order to restrain or reduce the harm of icing on atmospheric structures more effectively, there is an urgent need to find some new ways to prevent the surfaces of atmospheric structures from icing.

Normally, de-/anti-icing methods can be divided into two categories: active and passive. The active methods are currently widely used, which include mechanical scraping, electrothermal methods, and utilizing de-icing fluids [5]. Nevertheless, these methods are often expensive due to high energy requirements. The passive solutions mainly include various anti-icing coatings that prevent ice from adhering or becoming susceptible to detaching by natural airflow or solar radiation. Over the past decade, great efforts have been made to develop suitable anti-icing coatings and to understand the mechanism of ice formation. It has been found that the anti-icing coatings can repel coming cold water droplets, inhibit or delay ice formation, or reduce the ice adhesion strength [6–8].

Although the existing anti-icing coating cannot completely prevent the surface of structures from icing under harsh environmental conditions, the anti-ice coating is one of the most promising anti-icing methods, which can reduce the adhesion force of ice and accelerate the shedding of ice, thus alleviating the damage of ice coating. Since icing on exposed surface is inevitable in harsh environments, reducing ice adhesion is regarded as one of the most promising methods to mitigate the hazard from ice accretion [8,9]. The ice adhesion strength on aluminum surfaces reaches up to 600 kPa, while on icephobic coatings it can be lowered to 20 kPa or even lower [8]. That is, the icephobic coatings can reduce the ice adhesion strength by up to 95%. In this perspective, we mainly focused on the surface with low ice adhesion strength. The surfaces with ice adhesion strength lower than 100 kPa are defined as icephobic surfaces [9]. Over the last two decades, a number of anti-icing coatings have been developed [8,10–12]. However, due to process problems in production, convenience and high price, current coatings are still far from large-scale practical applications. Hence, a perspective is urgent to spur the development of anti-icing coatings towards commercialization.

Herein, we firstly recap the state-of-the-art hydrophobic coatings for anti-icing and then point out the deficiency and limitations of the current coatings. Afterwards, perspectives of future research directions are provided. Based on the wettability of coatings, the current anti-icing coatings can be divided into three categories: hydrophilic, hydrophobic, and superhydrophobic coatings. Herein, this paper focuses on hydrophobic coatings and superhydrophobic coatings, as shown in Figure 1.

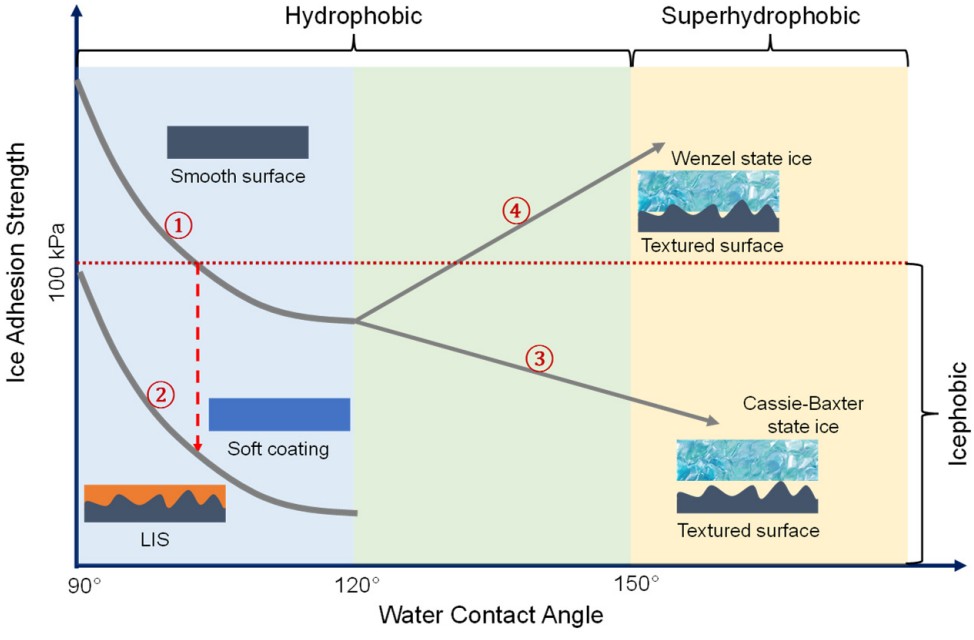

**Figure 1.** The schematic of hydrophobic and superhydrophobic coating design for anti-icing.

## 2. Hydrophobic Coatings

Hydrophobic coatings are thought to be one of the candidates for anti-icing due to their water repellency. Meuler et al. [13] described the relationship between ice adhesion strength and the wettability of smooth surfaces (Line ① in Figure 1). It was found that the ice adhesion strength is in proportion to $(1 + \cos \theta_{rec})$, where $\theta_{rec}$ is the receding water contact angle. Hence, many researchers devote themselves to lowering the surface energy of coatings [14–19]. Polysiloxane- and fluorine-containing coatings are usually used for achieving ultra-low surface energy, as shown in Figure 2 [8,18,20,21]. However, 120° is the limiting advancing contact angle on smooth surfaces. It means that there is a lower bound of ice adhesion strength by solely controlling the surface energy of the surface [8].

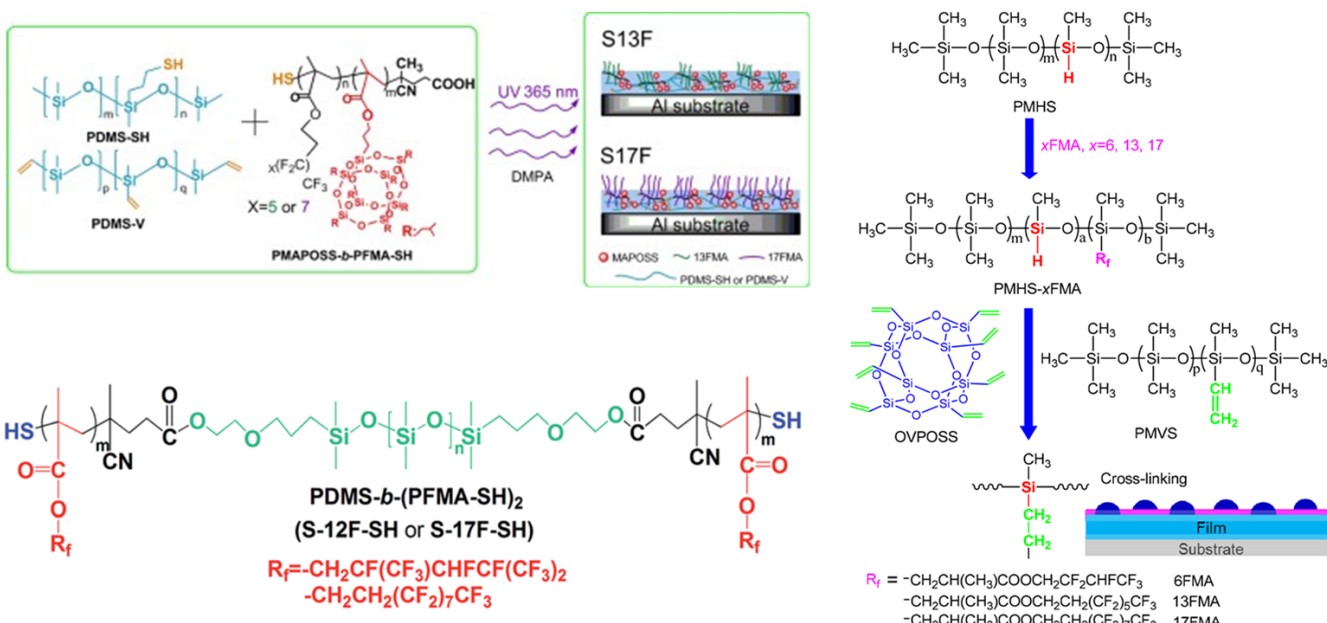

**Figure 2.** Polysiloxane- and fluorine-containing coatings for icephobicity [18,20,21]. Figures reproduced corresponding publishers Elsevier, RSC.

Fortunately, not only the wettability but also the elastic modulus of a coating can affect the ice adhesion strength [9,22]. By lowering the elastic modulus of coatings, the ice adhesion strength can decrease dramatically to an ultra-low value (<20 kPa), as shown in Line ② in Figure 1. Beemer et al. [22] have developed polydimethylsiloxane (PDMS) gels, which can offer ultra-low ice adhesion strength. Such ultra-low ice adhesion results from the low surface energy of PDMS as well as the low modulus of PDMS gel. The prepared PDMS gels exhibited an elastic modulus below 1 MPa, while ice shows an elastic modulus 2–3 order of magnitudes higher [8]. Such a tremendous modulus mismatch between PDMS gel and ice leads to the deformation incompatibility during the de-icing process, which will result in the formation of voids at the interface. The generated voids then serve as crack initiators to promote the removal of ice. Coincidentally, Golovin et al. [9] developed a series of durable icephobic materials with ultra-low ice adhesion strength by tuning the modulus and enabling interface slippage. Furthermore, under the guidance of classical fracture mechanics, Zhang et al. prepared a series of PDMS coating with sub-structures under the surface, which can render the deformation mismatch inside the coating itself and thus promote the crack generation at the interface [23,24]. Such macro-crack initiators (MACI) thus can lead to ultra-low ice adhesion strength. By using a similar design philosophy, Irajizad et al. prepared stress-localized durable icephobic surfaces that can achieve ice adhesion strength in the order of 1 kPa. Although these soft coatings present ultra-low ice adhesion strength and can survive after multiple icing/de-icing processes, they highly rely on the ultra-low modulus, which make the coating vulnerable. To address this problem and enhance the durability of soft coatings, Zhuo et al. have introduced a self-healing function [25,26] and a molecular pully (mechanical tuning) into soft coatings [27]. However, the easy deformability still limits the application scenarios of these coatings.

## 3. Superhydrophobic Coatings

The other option is introducing surface structure to break through the limitation of WCA. The superhydrophobic surfaces originate from lotus leaves and have a contact angle greater than 150° and a sliding angle lower than 10° [28]. The superhydrophobic surface has a micro-/nanoscale rough structure that can trap air firmly between the water droplets and substrate and minimize the water–solid contact area. Such a kind of water state is named the Cassie-Baxter state, and it enables the water repellency [28,29]. The trapped air

cushion layer at the solid–liquid interface can also serve as a heat conduction barrier to the droplets, which can delay the ice formation [30,31]. Furthermore, thanks to the air cushion, the real contact area between the superhydrophobic surface and the ice will be very low, as the Cassie-Baxter state water freezes into ice. The presence of an air cushion can also serve as a microcrack initiator to promote the interfacial fracture between the ice and the surface [32]. Hence, the ice adhesion strength of Cassie-Baxter state ice can be dramatically decreased, as shown in Line ③ in Figure 1.

Owing to the above properties, many scholars have studied anti-icing performance [33,34]. Pan et al. designed a new three-scale micro-/nanostructured superhydrophobic surface by combining ultrafine laser ablation and chemical oxidation [35], which demonstrated excellent anti-icing property. The surface is covered with densely grown nanograss and dispersed periodic microcone arrays. This surface exhibits excellent Cassie stability with a critical Laplacian pressure of up to 1450 Pa, which can prevent the penetration of water droplets into the microstructure and is essential for good anti-icing properties. The ice adhesion strength can be as low as 1.7 kPa. In addition, the ice adhesion strength can still remain below 10 kPa even after 10 cycles of the de-icing test, showing remarkable de-icing robustness. The dramatically decreased ice adhesion strength mainly results from the low real contact area and the presence of the microcrack initiator due to the surface structure [32]. However, microdroplets may form inside of the microstructure in high humidity environments, leading to the formation of Wenzel state water and the dysfunction of super hydrophobicity [36]. In addition, high speed supercooled microdroplets may also penetrate into the microstructure [37]. Once the microdroplets inside the structure form ice crystals, the surfaces will become more prone to freezing [38,39]. Furthermore, since the ice forms inside the structure (Wenzel state ice), mechanical interlocking will occur between the ice and superhydrophobic surfaces, thus sharply increasing the ice adhesion strength. The other issue is that the surface structure may be destroyed during de-icing or other mechanical loading, thus rendering the loss of super hydrophobicity and icephobicity. As shown in Figure 3b, Kulinich et al. have found that the ice-repellent properties become worse during icing/de-icing cycles [39]. Varanasi et al. also have reported that superhydrophobic surfaces can reduce ice adhesion to less than 100 kPa, but it expands to 200 kPa after icing/de-icing cycles [40].

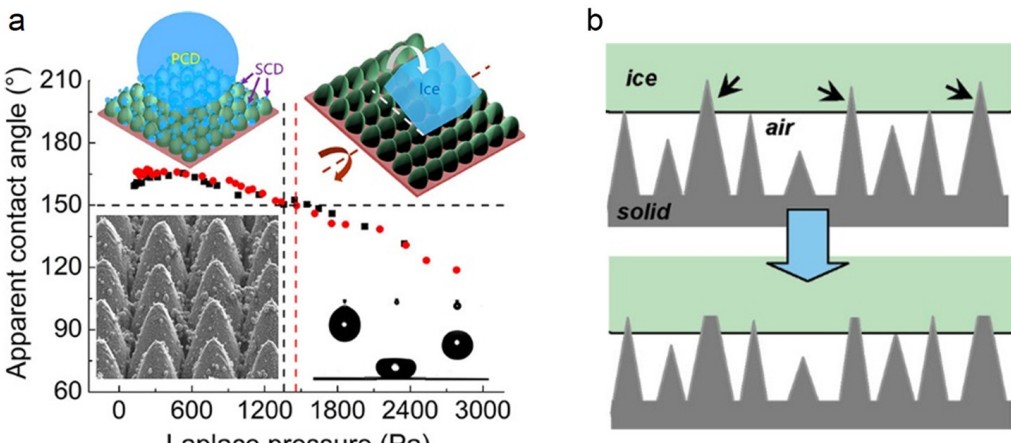

**Figure 3.** Superhydrophobic surfaces for anti-icing. (**a**) Three-scale micro-/nanostructured superhydrophobic surface [35]. (**b**) Microstructure of superhydrophobic surface is destroyed during removal of ice, resulting in a decrease of icephobicity [39]. Figures reproduced corresponding to publishers ACS.

Hu et al. synthesized particles with raspberry-like structures, and further formed a superhydrophobic and lipophilic surface [41]. The water contact angle of the coating was $152° \pm 2°$, and the oil contact angle was close to $0°$. The hydrophobic properties of composite particles can be adjusted by changing their surface morphology. Peng et al. prepared a stable superhydrophobic surface using a biological template replication method. Lotus leaves and rice were used as biological templates to construct artificial superhydrophobic surfaces [42].

Zheng et al. prepared a durable superhydrophobic coating by simply spraying graphene oxide/$ZrO_2$/PPS paint. This coating demonstrated good stability, which maintained a clean surface even after being repeatedly pulled in muddy water 100 times [43]. Xie et al. prepared epoxy/graphene oxide/$SiO_2$ multifunctional coatings on Mg-Zn-Ca alloys by a rotational spray-assisted self-assembly technique. The epoxy acted as a bonding layer, which could effectively enhance the mechanical robustness [44].

Besides structural destruction, the degradation of low surface energy materials is another crucial factor. Kulinich et al. systematically studied how the hydrophobicity of alkylsilane layers from octadecyltrimethoxysilane (ODTMS) changed over time after immersion in water [14]. As all the samples exhibited a decrease in their hydrophobicity over time, it is concluded that ODTMS layers are hydrolyzed gradually and slowly degrade. Even though the tests in the above work were conducted in water, there is little doubt that there will be pretty much the same reactions when/if silane-coated surfaces contact ice and/or overcooled water. As organo-silanes are still widely used as hydrophobic layers for anti-ice materials, how to avoid the degradation of silane-coated surfaces would be a crucial question.

### 4. Perspectives

As shown in Table 1, we have compared the adhesion performance of superhydrophobic coatings prepared from different materials. Many current hydrophobic coatings can reduce the ice adhesion strength to an ultra-low value (20 kPa), while the ice adhesion strength on bare substrates, e.g., aluminum, steel, and glass surfaces, is higher than 600 kPa [45]. Such a tremendous reduction in ice adhesion strength can promote the shedding of covered ice, thus showing great promise for anti-icing. However, many problems still limit the wide practical application of current anti-icing coatings [45].

**Table 1.** Comparison of ice adhesion strength.

| Material | Ice Adhesion Strength (kPa) at $-10\,^{\circ}$C | Ref. |
|---|:---:|:---:|
| Plasma polymerized hexamethyldisiloxane coating on aluminum | $100 \pm 25$ | [44] |
| Glass coated with hydrophobic nanoparticles and fluoroalkyl silane | $75 \pm 19$ | [44] |
| SLIPS-Al | 15 | [24] |
| LIT PDMS | 4 | [35] |
| A triple-scale MNGF superhydrophobic surface | 1.7 | [40] |

Although the ice adhesion strength of coatings can be lowered to 10 kPa or even Pa level, the lifetime, i.e., the mechanical durability of them, is unsatisfactory [46]. Many researchers are trying to find a solution to the mechanical durability of coatings by using different methods. The mechanical durability of coatings usually depends on the mechanical properties of coating materials, the structure of coatings, and the adhesion between coating and substrate (which is ignored by most researchers). The mechanical properties of coating materials relate to their microphase structure, which can be tuned by the molecular design. It means chemical structure design is an effective strategy for solving the mechanical properties of coatings. Zhuo et al. have demonstrated that by introducing a molecular pully and self-healing function, the mechanical durability can be dramatically improved [18,21,22]. More work can be focused in this direction. Structure design is also one way to improve the durability, especially for superhydrophobic surfaces. Deng et al. have designed a superhydrophobic surface with an armor-like structure [47], which can protect the nanostructures from the abrasion, thus providing mechanical durability. Such a kind of structure design can also be applied to the icephobic surface design towards mechanical robustness. The adhesion between coating and substrate is also one of the important parameters that will affect the durability of anti-icing coatings, since the

de-icing process usually requires shear loading, which may result in delamination when the adhesion between coating and substrate is not strong enough. The methods to increase the adhesion strength between coatings and substrate can be categorized into introducing covalent bonds, introducing physical interaction, and introducing intermedia layers. For example, silane coupling agents are widely used for enhancing the adhesion between organic coatings and inorganic substrates because they can form covalent bonds with both. Mussel-inspired chemistry is another example that can make the organic coatings strongly adhere to inorganic surfaces. Hydrogen bonds and metal-ligand coordination, as physical interactions, can also improve the interface strength. In the coating industry, tie-coating, which serves as the intermedia layer between top functional coating and substrate, is one of the most used methods to improve the adhesion between two layers. The adhesion between coating and substrate is of great importance, but few works focus on this for anti-icing coatings [48].

Functional superhydrophobic coatings such as superhydrophobic coatings with self-healing functions are gradually coming into the field of vision of researchers. The self-healing superhydrophobic coating can recover its superhydrophobic properties automatically after being damaged by external forces. In nature, organisms with superhydrophobicity, such as lotus leaves, can grow and construct micro-nano composite structures and secrete low surface energy substances to maintain their superhydrophobicity [49]. In recent years, superhydrophobic coatings with self-healing functions have attracted the attention of researchers. Researchers have tried to simulate the self-healing mechanism of animals and plants to improve the practicality and service life of superhydrophobic coatings [50,51]. At present, it is still impossible to realize true self-healing. The research on the mechanism of biological self-healing ability is the basis of preparing self-healing superhydrophobic coatings. In addition, compared with organic chemical materials, the micro-nano superhydrophobic surface structures constructed on the surface of metals or inorganic materials by laser etching and other methods have better stability; this method also has good application prospects [35,52]. However, its high manufacturing cost and complex production technology limit its large-scale engineering application at present.

Anti-icing coatings should be applied in different environment. Superhydrophobic surfaces can repel the coming cold-water droplets, therefore mitigating the ice accretion. However, such anti-icing properties may be lost in humid atmospheres [53]. Hence, the reliability of anti-icing surfaces in different environments, especially superhydrophobic surfaces, should be improved.

Ice accretion can occur on the surfaces of power lines, airplanes, solar panels, wind turbines and other infrastructures and transportation [54]. Various surfaces are composed with different materials, shapes, and sizes. It is therefore almost impossible to provide a universal coating which can work in different scenarios. Typically, the performance of an ice-repellent coating is evaluated by measuring the force $F$ to detach a specific area $A$ as well as the adhesion strength ($\tau_{ice} = F/A$). Many engineering structures (such as aircraft wings, wind turbine blades and ship hulls) have surface areas approaching thousands of square meters [35]. Thus, even with extremely icephobic coatings, structures with large surface areas require very high forces to separate the entire ice sheet from the surface. Then, Kevin et al. developed a new kind of low interfacial toughness material for large-area de-icing [9] When the interface length is relatively small, the shear strength of the interface $\hat{\tau}$ controls delamination and the ice rupture spontaneously along the entire interface; when the interface length is relatively large, interfacial toughness $\Gamma$ controls delamination, which leads to the generation and propagation of interfacial cracks. Thus, there is a critical bonded length at which a transition between the two modes of failure occurs, given by

$$L_C = \sqrt{2E_{ice}\Gamma h/\hat{\tau}^2} \tag{1}$$

where: $E_{ice}$ is the elastic modulus of ice with thickness h; $\Gamma$ is interfacial toughness; $\hat{\tau}$ is the ice adhesion strength value, $\hat{\tau}^2 = \tau_{ice}$. When $L > L_C$, the force required to separate the

ice is constant, regardless of the interface size. In addition, the detachment of ice from the surface is actually an interface fracture, whose mode will also be affected by the surface shape. As a result, various surfaces with different sizes and shapes should require different anti-icing coatings.

Furthermore, $\Gamma$ represents the energy required to de-bond an interfacial area and is equal to the area under the traction–separation curve. With a compliant coating between the ice and substrate, the deformation of the coating can be considered an integral part of the traction–separation curve. In this way, one can treat a coating as part of the interphase between the ice and the underlying substrate and model the entire interphase with a cohesive-zone analysis [55]. Then, the interfacial toughness can then be approximated as [9]

$$\Gamma \approx \hat{\tau}^2 t / 2G \tag{2}$$

where $G$ is the shear modulus, $t$ is the thickness of the coating, $\hat{\tau}^2 = \tau_{ice}$.

As discussed previously, the practical applications of anti-icing coatings are usually in different complex environments, which may cause different ice types (glaze, snow, frost, rime, ice) [41]. The formation processes of these types of ice are distinct. In addition, the adhesion strength of ice with different types shows different values [52–54,56,57]. Therefore, the study of the law of icing in different environments is indeed urgent. A cold room with controllable temperature, humidity, and water droplet size can facilitate the study of the law of icing. In addition, more wild platforms, such as the Xuefeng Mountain Energy Equipment Safety National Observation and Research Station, may be required for icing study.

## 5. Conclusions and Prospects

Icing can adversely affect the activities of daily living. The current approaches are to prevent icing and eliminate ice accumulation. In this paper, hydrophobic anti-icing coatings with important research significance were introduced. Although most of the current anti-icing coatings cannot inhibit ice growth, they can effectively decrease the ice adhesion strength and thus dramatically mitigate ice accretion, showing great promise. However, all of the current coatings need improved durability and reliability. It is also necessary to solve the technological problems in production and convenience, and the price problems in engineering applications.

For the research of anti-icing coatings, we shall not only aim at power grids, but also aircraft, high-speed trains, wind turbines and other relevant fields. It is almost impossible to provide a universal anti-icing coating that can work in different scenarios. The coating design should concern the size and shape of the surface and the corresponding application environment. To guide and facilitate the coating design, the study of the law of icing is urgent. For example, the icing on the surfaces of different infrastructures and equipment in different weather and environments. Therefore, to meet the technical requirements of large areas and the efficient de-icing of the surface of atmospheric structures, there is still a lot of work to be done before real large-scale engineering applications.

**Author Contributions:** Conceptualization, X.J.; methodology, M.Y., Y.L. and B.C.; validation, Y.Z. and P.W.; writing—original draft preparation, M.Y. and P.W.; writing—review and editing, X.J. and Y.Z.; supervision, X.J.; project administration, X.J.; funding acquisition, X.J. All authors have read and agreed to the published version of the manuscript.

**Funding:** The authors acknowledge the support from National Natural Science Foundation of China (NSFC) (Grant Nos. 51637002, 51977016).

**Institutional Review Board Statement:** Not applicable.

**Informed Consent Statement:** Not applicable.

**Data Availability Statement:** Not applicable.

**Conflicts of Interest:** The authors declare no conflict of interest.

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
