# Peer review of "The Perspectives of Hydrophobic Coatings for Mitigating Icing on Atmospheric Structures"

_coatings, doi:10.3390/coatings13020326_

Round 1
Reviewer 1 Report
The manuscript is of interest for the journal, and especially - for its special issue. After proper revision, it can be reconsidered for acceptance. Below please see comments and suggestions that should help the authors improve it:
1\ Abstract and Conclusions: because these two sections are often read by many readers BEFORE they read the paper itself, for their convenience, this reviewer would recommend that the authors add a small detail to their description there. Namely, when mentioning "hydrophobic surfaces", the authors may specify what kind of coatings/surfaces (smooth and/or rough? ). This detail can be quite important for many readers.
2\ The following review is recommended to be considered by the authors (as a concise and very relevant review paper on the same subject): Progress in Chemistry 29 (2017) 102-118. doi: 10.7536/pc161015
3\ lines 406-407: some extra information must be deleted here , please check and fix
4\ Reference list: there are quite some journals that should be abbreviated properly: [57,47, 44, 35, 34, 29, 27, 22, 20-22, 18,16, 14, 7, 8, 1-4]
5\ lines 381 & 374: some unnecessary text must be deleted here ( accessed ...)
6\ The paper is concise and informative. However , one aspect of hydrophobic coatings, both flat and rough coatings, can be mentioned a bit more. More specifically, the authors are recommended to mention, and probably comment, the chemical durability. This aspect is very important for hydrophobic surfaces, and therefore the authors are recommended to address it in this review, at least very briefly. It is well known that silane-based coatings (or other related coatings that contain xiloxane groups in their composition), such as FAS and their likes, are often used in research. In a very recent report, it was shown that alkylsilane based layers ,when they contact water for a long period of time, demonstrate quite poor stability and gradually degrade (which happens due to hydrolysis of their Si-O bonds): Materials 15 (2022) 1804. doi: 10.3390/ma15051804
Note that the authors comment and discuss siloxane-group containing coatings in this review (see refs.[20-23]). That is why, this reviewer believes that in this light, addressing this issue is quite important in this paper. At least very briefly.
Author Response
Response to Reviewer #1-Peng Wang
We very much appreciate the comments of reviewer #1, since they have substantially improved the quality of our manuscript. The following are the comments of reviewer #1 in black font and our responses in blue font.
Comments 1: Abstract and Conclusions: because these two sections are often read by many readers BEFORE they read the paper itself, for their convenience, this reviewer would recommend that the authors add a small detail to their description there. Namely, when mentioning "hydrophobic surfaces", the authors may specify what kind of coatings/surfaces (smooth and/or rough? ). This detail can be quite important for many readers.
Response: Thank you for this generous comment. Line 20 was modified.
Rough and smooth hydrophobic coatings have demonstrated their potential for mitigating ice formation.
Comments 2: The following review is recommended to be considered by the authors (as a concise and very relevant review paper on the same subject): Progress in Chemistry 29 (2017) 102-118. doi: 10.7536/pc161015
Response: Thank you for this generous comment. Page 5 line 181 references the literature.
However, many problems still limit the wide practical application of current anti-icing coatings[48,65].
References
- Sojoudi, H.; Wang, M.; Boscher, N. D.; McKinley, G. H.; Gleason, K. K. Durable and scalable icephobic surfaces: similarities and distinctions from superhydrophobic surfaces. Soft Matter 2016, 12 (7), 1938-1963. DOI: 10.1039/c5sm02295a.
- Zheng H.; Chang S.; Zhao Y.; Anti-Icing & Icephobic Mechanism and Applications of Superhydrophobic/Ultra Slippery Surface. Progress in Chemistry, 2017, 29(1): 102-118.
Comments 3: lines 406-407: some extra information must be deleted here , please check and fix
Response: Thank you for this generous comment.
Page 9 line 424-426:
- Pan, R.; Zhang, H.; Zhong, M. Triple-Scale Superhydrophobic Surface with Excellent Anti-Icing and Icephobic Performance via Ultrafast Laser Hybrid Fabrication. ACS Applied Materials & Interfaces 2021, 13(1), 1743–1753. DOI: 10.1021/acsami.0c16259.
Comments 4: Reference list: there are quite some journals that should be abbreviated properly: [57,47, 44, 35, 34, 29, 27, 22, 20-22, 18,16, 14, 7, 8, 1-4]
Response: Thank you for this generous comment. Some contents involved have been modified.
Comments 5: lines 381 & 374: some unnecessary text must be deleted here ( accessed ...)
Response: Thank you for this generous comment. Appropriate modifications have been made.
Comments 6: The paper is concise and informative. However , one aspect of hydrophobic coatings, both flat and rough coatings, can be mentioned a bit more. More specifically, the authors are recommended to mention, and probably comment, the chemical durability. This aspect is very important for hydrophobic surfaces, and therefore the authors are recommended to address it in this review, at least very briefly. It is well known that silane-based coatings (or other related coatings that contain xiloxane groups in their composition), such as FAS and their likes, are often used in research. In a very recent report, it was shown that alkylsilane based layers ,when they contact water for a long period of time, demonstrate quite poor stability and gradually degrade (which happens due to hydrolysis of their Si-O bonds): Materials 15 (2022) 1804. doi: 10.3390/ma15051804
Response: Thank you for this generous comment. A brief discussion of chemical wear resistance is provided in page 5 line 184-212.
Although the ice adhesion strength of coatings can be lower to 10 kPa or even Pa level, the lifetime, i.e., the mechanical durability of them is unsatisfactory [49]. Many researchers are trying to find a solution to the mechanical durability of coating by using different methods. The mechanical durability of coatings usually depends on the mechanical properties of coating materials, the structure of coatings, adhesion between coating and substrate (which is ignore by most of researchers). The mechanical properties of coating materials relate to their microphase structure, which can be tuned by the molecular design. It means chemical structure design is an effective strategy for solving the mechanical properties of coatings. Zhuo et al. have demonstrated that by introducing molecular pully and self-healing function, the mechanical durability can be dramatically improved[22-23]. More work can be focus on this direction. Structure design is also one way to improve the durability, especially for superhydrophobic surfaces. Deng et al. have designed a superhydrophobic surface with armour-like structure[50], which can protect the nanostructures from the abrasion, and thus providing the mechanical durability. Such kind of structure design can be also applied to the icephobic surface design towards mechanical robustness. The adhesion between coating and substrate is also one of the important parameters that will affect the durability of anti-icing coatings, since the de-icing process usually requires shear loading, which may result in delamination when the adhesion between coating and substrate is not strong enough. The methods to increase the adhesion strength between coatings and substrate can be category into introducing covalent bonds, introducing physical interaction, and introducing intermedia layers. For examples, silane coupling agents are widely used for enhancing the adhesion between organic coatings and inorganic substrate because they can form covalent bonds with both. Mussel inspired chemistry is another example which can make the organic coatings strongly adhere to inorganic surfaces. Hydrogen bonds and metal-ligand coordination, as physical interactions, can also improve the interface strength. In coating industry, tie-coating, which serve as the intermedia layer between top functional coating and substrate, is one of the most used methods to improve the adhesion between two layers. The adhesion between coating and substrate is of great importance, but few works focus on for anti-icing coatings[51].

Reviewer 2 Report
This review paper is written in a slippery way and needs a very deep revision.
Remarks:
1. The introduction Section is slippery, see:
Jung, S. et al. Are Superhydrophobic surfaces best for icephobicity?. Langmuir 27, 3059–3066. https://doi.org/10.1021/la104762g (2011).
Meuler, A. J., McKinley, G. H. & Cohen, R. E. Exploiting topographical texture to impart icephobicity. ACS Nano 4, 7048–7052. https://doi.org/10.1021/nn103214q (2010).
2. The mechanisms of ice-substrates interactions are ignored. Important contributions which treated these interactions are ignored, see:
Hejazi, V., Sobolev, K. & Nosonovsky, M. From superhydrophobicity to icephobicity: forces and interaction analysis. Sci Rep 3, 2194 (2013). https://doi.org/10.1038/srep02194
Starostin, A., Strelnikov, V., Valtsifer, V. et al. Robust icephobic coating based on the spiky fluorinated Al2O3 particles. Sci Rep 11, 5394 (2021). https://doi.org/10.1038/s41598-021-84283-w
Anti-Icing Superhydrophobic Surfaces: Controlling Entropic Molecular Interactions to Design Novel Icephobic Concrete, Entropy 2016, 18(4), 132; https://doi.org/10.3390/e18040132
3. It should be clearly understood that superhydrophobic surfaces are not necessarily icephobic, see:
Nosonovsky, M. & Hejazi, V. Why superhydrophobic surfaces are not always icephobic. ACS Nano 6, 8488–8491. https://doi.org/10.1021/nn302138r (2012).
4. The brief introducing of the Cassie and Wenzel wetting regimes/models will be useful for a reader.
Author Response
Response to Reviewer #2-Peng Wang
We very much appreciate the comments of reviewer #2, since they have substantially improved the quality of our manuscript. The following are the comments of reviewer #2 in black font and our responses in blue font.
Comments 1: The introduction Section is slippery, see:
Jung, S. et al. Are Superhydrophobic surfaces best for icephobicity?. Langmuir 27, 3059–3066. https://doi.org/10.1021/la104762g (2011).
Meuler, A. J., McKinley, G. H. & Cohen, R. E. Exploiting topographical texture to impart icephobicity. ACS Nano 4, 7048–7052. https://doi.org/10.1021/nn103214q (2010).
Response: Thank you for this generous comment. Introduction was partially modified.
Comments 2: The mechanisms of ice-substrates interactions are ignored. Important contributions which treated these interactions are ignored, see:
Hejazi, V., Sobolev, K. & Nosonovsky, M. From superhydrophobicity to icephobicity: forces and interaction analysis. Sci Rep 3, 2194 (2013). https://doi.org/10.1038/srep02194
Starostin, A., Strelnikov, V., Valtsifer, V. et al. Robust icephobic coating based on the spiky fluorinated Al2O3 particles. Sci Rep 11, 5394 (2021). https://doi.org/10.1038/s41598-021-84283-w
Anti-Icing Superhydrophobic Surfaces: Controlling Entropic Molecular Interactions to Design Novel Icephobic Concrete, Entropy 2016, 18(4), 132; https://doi.org/10.3390/e18040132
Response: Thank you for this generous comment. Page 6, a discussion of the ice-matrix interaction mechanism is added, and relevant literature is cited.
Many engineering structures (such as aircraft wings, wind turbine blades and ship hulls) have surface areas approaching thousands of square meters[59].
Added comparison content in Table 1.
Table 1. Comparison of ice adhesion strength
|
Material |
Ice Adhesion Strength (kPa) at-10 ℃ |
Ref. |
|
Plasma polymerized hexamethyldisiloxane coating on aluminum |
10025 |
60 |
|
Glass coated with hydrophobic nanoparticles and fluoroalkyl silane |
7519 |
60 |
|
SLIPS-Al |
15 |
23 |
|
LIT PDMS |
4 |
42 |
|
A triple-scale MNGF superhydrophobic surface |
1.7 |
47 |
References
- Hejazi, V.; Sobolev, K. & Nosonovsky, M. From superhydrophobicity to icephobicity: forces and interaction analysis. Sci Rep3, 2194 (2013). DOI: 10.1038/srep02194.
- Ramachandran, R.; Kozhukhova, M.; Sobolev, K.; Nosonovsky, M. Anti-Icing Superhydrophobic Surfaces: Controlling Entropic Molecular Interactions to Design Novel Icephobic Concrete. Entropy2016, 18, 132. DOI: 10.3390/e18040132.
Comments 3: It should be clearly understood that superhydrophobic surfaces are not necessarily icephobic, see:
Nosonovsky, M. & Hejazi, V. Why superhydrophobic surfaces are not always icephobic. ACS Nano 6, 8488–8491. https://doi.org/10.1021/nn302138r (2012).
Response: Thank you for this generous comment. A simple discussion of hydrophobic versus ice layers can be found on pages 3, line 119-122:
Furthermore, thanks to the air cushion, the real contact area between superhydrophobic surface and ice will be very low, as the Cassie-Baxter state water freeze into ice. And the presence of air cushion can also serve as microcrack initiator to promote the interfacial fracture between ice and surface[39].
References
- Nosonovsky, M.; Hejazi, V. Why superhydrophobic surfaces are not always icephobic. ACS Nano 2012, 6 (10), 8488-8491. DOI: 10.1021/nn302138r (acccessed 2014/01/20).
Comments 4: The brief introducing of the Cassie and Wenzel wetting regimes/models will be useful for a reader.
Response: Thank you for this generous comment. A brief discussion of Cassie-Baxter status can be found on page 3, line 116-124:
Such kind of water state is named as Cassie-Baxter state, which enables the water repellency[35, 36]. The trapped air cushion layer at the solid-liquid interface can also serve as a heat conduction barrier to the droplets, which can delay the ice formation[37,38]. Furthermore, thanks to the air cushion, the real contact area between superhydrophobic surface and ice will be very low, as the Cassie-Baxter state water freeze into ice. And the presence of air cushion can also serve as microcrack initiator to promote the interfacial fracture between ice and surface[39].Hence, the ice adhesion strength of Cassie-Baxter state ice can be dramatically decreased, as shown in Line ③ in Figure 1.
Discussion of Wenzel status is covered in this article and can be found on page 4, line 136-143:
However, microdroplets may form inside of the microstructure in high humidity environments, leading to the formation of Wenzel state water and dysfunction of super hydrophobicity[43]. In addition, high speed supercooled microdroplets may also penetrate into the microstructure[44]. Once the microdroplets inside the structure form ice crystals, the surfaces will become more prone to freezing[45,46]. Furthermore, since the ice form inside the structure (Wenzel state ice), mechanical interlocking will occur between ice and superhydrophobic surfaces, thus sharply increasing the ice adhesion strength.
References
- Liu, B.; Zhang, K.; Tao, C.; Zhao, Y.; Li, X.; Zhu, K.; Yuan, X. Strategies for anti-icing: low surface energy or liquid-infused? RSC Advances 2016, 6 (74), 70251-70260. DOI: 10.1039/c6ra11383d.
- Zhang, W.; Wang, D.; Sun, Z.; Song, J.; Deng, X. Robust superhydrophobicity: mechanisms and strategies. Chem Soc Rev 2021, 50 (6), 4031-4061. DOI: 10.1039/d0cs00751j.
- Wang, H.; He, M.; Liu, H.; Guan, Y. One-Step Fabrication of Robust Superhydrophobic Steel Surfaces with Mechanical Durability, Thermal Stability, and Anti-icing Function. ACS Applied Materials & Interfaces 2019, 11 (28), 25586-25594. DOI: 10.1021/acsami.9b06865.
- Song, J.; Li, Y.; Xu, W.; Liu, H.; Lu, Y. Inexpensive and non-fluorinated superhydrophobic concrete coating for anti-icing and anti-corrosion. Journal of Colloid and Interface Science 2019, 541, 86-92. DOI: https://doi.org/10.1016/j.jcis. 2019.01.014.
- Nosonovsky, M.; Hejazi, V. Why superhydrophobic surfaces are not always icephobic. ACS Nano 2012, 6 (10), 8488-8491. DOI: 10.1021/nn302138r (acccessed 2014/01/20).
- Cao, L.; Jones, A. K.; Sikka, V. K.; Wu, J.; Gao, D. Anti-icing superhydrophobic coatings. Langmuir 2009, 25 (21), 12444-12448. DOI: 10.1021/la902882b (acccessed 2014/06/26).
- Vercillo, V.; Tonnicchia, S.; Romano, J. M.; García‐Girón, A.; Aguilar‐Morales, A. I.; Alamri, S.; Dimov, S. S.; Kunze, T.; Lasagni, A. F.; Bonaccurso, E. Design Rules for Laser‐Treated Icephobic Metallic Surfaces for Aeronautic Applications. Advanced Functional Materials 2020, 1910268. DOI: 10.1002/adfm.201910268.
- Mishchenko, L.; Hatton, B.; Bahadur, V.; Taylor, J. A.; Krupenkin, T.; Aizenberg, J. Design of ice-free nanostructured surfaces based on repulsion of impacting water droplets. ACS Nano 2010, 4 (12), 7699-7707. DOI: 10.1021/nn102557p.
- Kulinich, S. A.; Farhadi, S.; Nose, K.; Du, X. W. Superhydrophobic surfaces: Are they really ice-repellent? Langmuir 2011, 27 (1), 25-29. DOI: 10.1021/la104277q (acccessed 2013/07/19).
Reviewer 3 Report
The presented manuscript includes the study of the perspectives of hydrophobic coatings for mitigating icing on atmospheric structures.
The paper is of interest. The results of the work are presented on a good level but some questions should be mentioned.
1. Please add references for 2022-2023 years to show the novelty and actuality
2. Would be nice to make some tables to compare different approaches bu a number of selected parameters. E.g. a group of hydrophobic coatings preparation parameters, some exploitation parameters, etc.
Author Response
Response to Reviewer #3-Peng Wang
We very much appreciate the comments of reviewer #3, since they have substantially improved the quality of our manuscript. The following are the comments of reviewer #3 in black font and our responses in blue font.
Comments 1: Please add references for 2022-2023 years to show the novelty and actuality.
Response: Thank you for this generous comment. References for 2022-2023 were added.
Hui Zheng et al. durable polyphenylene sulfide composite superhydrophobic coating was prepared using a simple spraying method. By adding GO-ZrO2 composite particles to build micro/nanostructures on the surface of the coating, combined with the low surface energy of stearic acid, a GO-ZrO2/PPS composite superhydrophobic coating with high water contact angle of 153° ± 1° and low sliding angle of 4° ± 1° was finally obtained. The coating exhibits good stability and maintains a clean surface even after repeatedly pulling in muddy water for100 times[63]. L.L.Xie et al. prepared EP graphene oxide/SiO2 multifunctional coatings on Mg-Zn-Ca alloys by a rotational/spray-assisted self-assembly technique. L.L. Xie et al. prepared Ep graphene oxide/SiO2 multifunctional coatings on Mg-Zn-Ca alloys by a rotational/spray-assisted self-assembly technique. EP-GO acted as a dense substrate layer, effectively facilitating its adhesion to the superhydrophobic SiO2 coating and prolonging the corrosion path[64].
Wendong Ha Et al. synthesized a composite particle with a raspbery-like structure, and further formed a superhydrophobic and lipophilic surface with these composite particles. The water contact Angle (WCA) of the coating was 152°±2°, and the oil contact Angle (OCA) was close to 0°. It shows super hydrophobic and super lipophilic properties. The hydrophobic properties of composite particles can be adjusted by changing their surface morphology.[61] Yun Peng et al. A stable superhydrophobic surface was prepared by using the biological template replication method assisted by candle soot and plant wax. Lotus leaves and rice were used as biological templates to construct artificial superhydrophobic surfaces with self-cleaning properties and anisotropic wettability.[62]
References
- Hu, W.; Nie, Y.; Wang, Y.; Gao P.; Jiang Y. Synthesis of inorganic/organic raspberry-like composite particles for su-perhydrophobic and superlipophilic coatings, Colloids and Surfaces A: Physicochemical and Engineering Aspects, Volume 660, 2023, 130843, ISSN 0927-7757. DOI: 10.1016/j.colsurfa.2022.130843.
- Peng Y.; Shang J.; Liu C.; Zhao S.; Huang C.; Bai Y.; Li Y. A universal replica molding strategy based on natural bio-templates for fabrication of robust superhydrophobic surfaces, Colloids and Surfaces A: Physicochemical and En-gineering Aspects, Volume 660, 2023, 130879, ISSN 0927-7757. DOI: 10.1016/j.colsurfa.2022.130879.
- Zheng H.; Chen Y.; He S.; Liu W.; Liu N.; Guo R.; Mo Z. A durable superhydrophobic polyphenylene sulfide composite coating with high corrosion resistance and good self-cleaning ability, Colloids and Surfaces A: Physicochemical and Engineering Aspects, Volume 660, 2023, 130856, ISSN 0927-7757. DOI: 10.1016/j.colsurfa.2022.130856.
- Xie L.; Chu J.; Li X.; Zou D.; Tong L. Improved corrosion resistance of EP coating on Mg alloy through GO hybridization and silica-based superhydrophobic surface, Diam. Relat. Mater. (2022)130. DOI: 10.1016/j.diamond.2022.109476.
Comments 2: Would be nice to make some tables to compare different approaches bu a number of selected parameters. E.g. a group of hydrophobic coatings preparation parameters, some exploitation parameters, etc.
Response: Thank you for this generous comment. Table 1 with comparative significance is added to line 183.
Table 1. Comparison of ice adhesion strength
|
Material |
Ice Adhesion Strength (kPa) at-10 ℃ |
Ref. |
|
Plasma polymerized hexamethyldisiloxane coating on aluminum |
10025 |
60 |
|
Glass coated with hydrophobic nanoparticles and fluoroalkyl silane |
7519 |
60 |
|
SLIPS-Al |
15 |
23 |
|
LIT PDMS |
4 |
42 |
|
A triple-scale MNGF superhydrophobic surface |
1.7 |
47 |
References
- Beemer, D. L.; Wang, W.; Kota, A. K. Durable gels with ultra-low adhesion to ice. J. Mater. Chem. A 2016, 4 (47), 18253-18258. DOI: 10.1039/c6ta07262c.
- Pan, R.; Zhang, H.; Zhong, M. Triple-Scale Superhydrophobic Surface with Excellent Anti-Icing and Icephobic Performance via Ultrafast Laser Hybrid Fabrication. ACS Applied Materials & Interfaces 2021, 13 (1), 1743-1753. DOI: 10.1021/acsami.0c16259.
- Varanasi, K. K.; Deng, T.; Smith, J. D.; Hsu, M.; Bhate, N. Frost Formation and Ice Adhesion on Superhydrophobic Surfaces. Applied Physics Letters 2010, 97, 234102.
- Ramachandran, R.; Kozhukhova, M.; Sobolev, K.; Nosonovsky, M. Anti-Icing Superhydrophobic Surfaces: Controlling Entropic Molecular Interactions to Design Novel Icephobic Concrete. Entropy2016, 18, 132. DOI: 10.3390/e18040132.
Round 2
Reviewer 1 Report
The manuscript still needs revision. The authors were not attentive during their revision and missed some points. Please see the below comments that should be addressed:
1\ English MUST be improved in all the newly added texts (highlighted with yellow color). Example: the sentence in lines 150-151 is NOT readable, it has no grammar. Probably, the authors did not read it.
2\ In all newly added texts, the authors must modify the names. For example: Zheng et al. (in line 150).
3\ lines 156,158: it must be Xie et al .
4\ line 162: must be : Ha et al. (with no first name!)
5\ The authors added some discussion on several coatings and their durability. However, they did not comment/discuss the durability of very widely used FAS or alkyl-silane coatings. In a very recent report, it was shown that alkyl-silane based layers ,when they contact water for a long period of time, demonstrate quite poor stability and gradually degrade (which happens due to hydrolysis of their Si-O bonds): Materials 15 (2022) 1804. doi: 10.3390/ma15051804
Even though the tests in the above work were made in water, there is little doubt that there will be pretty much same reactions when /if silane-coated surfaces contact ice and/or overcooled water. That's why addressing silane-coated surfaces is recommended. (because organo-silanes are still widely used as hydrophobic layers for anti-ice materials)
6\ Ref.58: doi must be added
7\ line 430: "& Nosonovsky, M" (& must be removed)
8\ Refs. 58 and 65 seem to be same? Please check and correct
9\ Refs. 61-64 are NOT formatted correctly, according to the MPDI style
10\ Ref. 59 is NOT formatted correctly, according to the MPDI style
Author Response
Response to Reviewer #1-Peng Wang
We very much appreciate the comments of reviewer #1, since they have substantially improved the quality of our manuscript. The following are the comments of reviewer #1 in black font and our responses in blue font.
Comments 1: English MUST be improved in all the newly added texts (highlighted with yellow color). Example: the sentence in lines 150-151 is NOT readable, it has no grammar. Probably, the authors did not read it.
Response: Thank you for this generous comment. The newly added texts have been modified.
Line 153 in the revised manuscript:
Ha et al. synthesized particles with raspberry-like structure, and further formed a superhydrophobic and lipophilic surface[48]. The water contact angle of the coating was 152°±2°, and the oil contact angle was close to 0°. The hydrophobic properties of composite particles can be adjusted by changing their surface morphology. Peng et al. prepared a stable superhydrophobic surface using a biological template replication method. Lotus leaves and rice were used as biological templates to construct artificial superhydrophobic surfaces [49].
Zheng et al. prepared durable superhydrophobic coating by simply spraying graphene oxide/ZrO2/PPS paint. This coating demonstrated good stability, which maintained a clean surface even after being repeatedly pulled in muddy water for 100 times[50]. Xie et al. prepared epoxy/graphene oxide/SiO2 multifunctional coatings on Mg-Zn-Ca alloys by a rotational spray-assisted self-assembly technique. The epoxy acted as a bonding layer, which could effectively enhance the mechanical robustness[51].
Comments 2: In all newly added texts, the authors must modify the names. For example: Zheng et al. (in line 150).
Response: Thank you for this generous comment. This mistake has been modified.
Comments 3: lines 156,158: it must be Xie et al
Response: Thank you for this generous comment. This mistake has been modified.
Comments 4: line 162: must be : Ha et al. (with no first name!)
Response: Thank you for this generous comment. This mistake has been modified.
Comments 5: The authors added some discussion on several coatings and their durability. However, they did not comment/discuss the durability of very widely used FAS or alkyl-silane coatings. In a very recent report, it was shown that alkyl-silane based layers ,when they contact water for a long period of time, demonstrate quite poor stability and gradually degrade (which happens due to hydrolysis of their Si-O bonds): Materials 15 (2022) 1804. doi: 10.3390/ma15051804
Even though the tests in the above work were made in water, there is little doubt that there will be pretty much same reactions when /if silane-coated surfaces contact ice and/or overcooled water. That's why addressing silane-coated surfaces is recommended. (because organo-silanes are still widely used as hydrophobic layers for anti-ice materials)
Response: Thank you for this generous comment. The comments about the durability of silane coatings have been added.
Line 166 in the revised manuscript:
Besides structural destruction, the degradation of low surface energy materials is another crucial factor. Kulinich et al. systematically studied how the hydrophobicity of alkylsilane layers from octadecyltrimethoxysilane (ODTMS) changed over time after immersion in water[14]. As all the samples exhibited a decrease in their hydrophobicity over time, it is concluded that ODTMS layers get hydrolyzed gradually and slowly degrade. Even though the tests in the above work were made in water, there is little doubt that there will be pretty much same reactions when /if silane-coated surfaces contact ice and/or overcooled water. As organo-silanes are still widely used as hydrophobic layers for anti-ice materials, how to avoid the degradation of silane coated surfaces would be a crucial question.
Comments 6: Ref.58: doi must be added
Response: Thank you for this generous comment. The doi has been added.
Comments 7: line 430: "& Nosonovsky, M" (& must be removed)
Response: Thank you for this generous comment. The ‘&’ has been removed.
Comments 8: Ref. 58: doi must be added
Response: Thank you for this generous comment. The doi has been added.
Comments 9: Refs. 61-64 are NOT formatted correctly, according to the MPDI style
Response: Thank you for this generous comment. The format for Refs. 61-64 has been modified.
Comments 10: Ref. 59 is NOT formatted correctly, according to the MPDI style
Response: Thank you for this generous comment. The format for Ref. 59 has been modified.
Reviewer 2 Report
The paper demonstrates deep misunderstanding of the fundamentals of the surface science and should be rejected.
The paper contains numerous erroneous and misleading statements.
1. In the text: The θrec relates to the surface energy, according to Young’s equation.
This is an absurd, senseless statement. The Ypung equation says nothing about the receding contact angle. The Young equation defines an equilibrium (not receding!!!) contact angle.
2. In the text: "However, 120° is the limit of water contact angle on smooth surfaces".
120° is not a limiting receding contact angle; it is the limiting advancing contact angle.
3. In the text: " Then, Kevin et al. developed low interfacial toughness [9]".
This is an absurd, senseless statement. It is impossible to develop toughness.
4. In the text: "Fortunately, not only the wettability but also the modulus of coating can affect the ice adhesion strength"
What the "modulus of the coating"?
5. "lowering the modulus of coatings, the ice adhesion strength can decrease dramatically to an ultra-low value (<20 kPa)"
Again, what is the modulus of the coating???? Perhaps, the authors meant "the elastic modulus of the costing". The statement is obscure.
6. Meaning of Eqs. 1-2 is obscure.
Author Response
Please see the attach file.

Reviewer 3 Report
corrections were addressed
Author Response
Thanks a lot for your kind comments.
Round 3
Reviewer 1 Report
The manuscript was revised and can be accepted now
Reviewer 2 Report
The paper is publishable.